# Engineering vanilloid-sensitivity into the rat TRPV2 channel

**Feng Zhang[1†], Sonya M Hanson[1,2†‡], Andres Jara-Oseguera[1], Dmitriy Krepkiy[1], Chanhyung Bae[1], Larry V Pearce[3], Peter M Blumberg[3], Simon Newstead[2], Kenton J Swartz[1]***

[1]Molecular Physiology and Biophysics Section, Porter Neuroscience Research Center, National Institute of Neurological Disorders and Stroke, National Institutes of Health, Bethesda, United States; [2]Department of Biochemistry, University of Oxford, Oxford, United Kingdom; [3]Laboratory of Cancer Biology and Genetics, Center for Cancer Research, National Cancer Institute, Bethesda, United States

**Abstract** The TRPV1 channel is a detector of noxious stimuli, including heat, acidosis, vanilloid compounds and lipids. The gating mechanisms of the related TRPV2 channel are poorly understood because selective high affinity ligands are not available, and the threshold for heat activation is extremely high (>50°C). Cryo-EM structures of TRPV1 and TRPV2 reveal that they adopt similar structures, and identify a putative vanilloid binding pocket near the internal side of TRPV1. Here we use biochemical and electrophysiological approaches to investigate the resiniferatoxin(RTx) binding site in TRPV1 and to explore the functional relationships between TRPV1 and TRPV2. Collectively, our results support the interaction of vanilloids with the proposed RTx binding pocket, and demonstrate an allosteric influence of a tarantula toxin on vanilloid binding. Moreover, we show that sensitivity to RTx can be engineered into TRPV2, demonstrating that the gating and permeation properties of this channel are similar to TRPV1.

***For correspondence:** swartzk@ninds.nih.gov

[†]These authors contributed equally to this work

**Present address:** [‡]Computational Biology Program, Memorial Sloan Kettering Cancer Center, New York, United States

## Introduction

Transient receptor potential (TRP) channels are a large family of cation selective channels, some of which function in sensory terminals to detect a wide array of different stimuli, including temperature, lipid messengers and natural products such as capsaicin, menthol, wasabi and mustard oil (*Ramsey et al., 2006*). The TRPV1 channel is one of the best studied TRP channels, and is activated by vanilloids like capsaicin from hot chili peppers, extracellular acid, the double-knot toxin (DkTx) from tarantula venom and noxious heat (*Bohlen and Julius, 2012*; *Julius, 2013*). Structures of TRPV1 have been solved by cryo-electron microscopy (EM) (*Cao et al., 2013*; *Liao et al., 2013*) (*Figure 1A*), revealing that the transmembrane domain of the TRP channel adopts a structure that is remarkably similar to voltage-activated potassium channels (*Long et al., 2007*), as well inositol trisphosphate and ryanodine receptors (*Yan et al., 2015*; *Zalk et al., 2015*). Protons and DkTx are known to activate TRPV1 from the external side of the pore; the residues involved in proton activation have been identified by mutagenesis (*Jordt et al., 2000*; *Ryu et al., 2003*; *Ryu et al., 2007*), and the DkTx binding site on the pore domain was initially localized by comparing apo and DkTx-bound cryo-EM structures of TRPV1 (*Cao et al., 2013*; *Liao et al., 2013*), and subsequently refined by docking the NMR structure of the toxin into the cryo-EM maps of TRPV1 (*Jara-Oseguera et al., 2016*). A Na[+] ion binding site within the external pore nearby to where DkTx binds was recently shown to stabilize the closed state of TRPV1, and to be allosterically coupled with temperature-sensor activation (*Jara-Oseguera et al., 2016*). Thus the external pore appears to be a critical nexus for allosteric gating in TRPV1. The cryo-EM structure of TRPV1 in the presence of resiniferatoxin (RTx)

**eLife digest** Ion channels form pores in cell membranes and can open and close to allow specific ions to flow from one side of the membrane to the other. Humans and other mammals rely on an ion channel protein called TRPV1 to sense heat, and this protein is activated by the active ingredient in hot chili peppers, a chemical called capsaicin that belongs to a group of compounds called vanilloids.

An ion channel called TRPV2 is related to TRPV1 and has a similar shape, but is less well understood. This is partly because there are no chemicals that are known to selectively activate this channel, and it is not activated by the vanilloids that activate TRPV1.

Zhang, Hanson et al. have now investigated where vanilloid compounds bind to activate the rat TRPV1 channel and if the binding site can be created in the rat TRPV2 as well. Like all proteins, these channels are built from smaller units called amino acids. The results from an array of approaches identified four specific amino acids that can be exchanged between TRPV1 and TRPV2 to swap their ability to bind and be activated by resiniferatoxin (a vanilloid isolated from a cactus-like plant called resin spurge). When rat TRPV2 channels were engineered to contain these four amino acids, resiniferatoxin could easily open the channel. In addition, the engineered channel allowed ions to pass through in a way that is similar to TRPV1, which suggests that the channels are more similar than previously believed.

Together the findings show how the TRPV2 channel can be made sensitive to vanilloids. This will help researchers to solve the structure of this ion channel when it is active and open. In addition, the findings may also allow the role of TRPV2 to be explored further in animals, such as mice.

revealed the presence of an additional density at the interface between the S1-S4 domains and the S5-S6 pore domains near the internal side of the membrane (*Figure 1B*) (*Cao et al., 2013*). Although this EM density is insufficient to identify where vanilloids bind, it is consistent with mutagenesis and computational studies (*Elokely et al., 2016*; *Gavva et al., 2004*; *Jordt and Julius, 2002*).

In contrast to TRPV1, relatively little is known about the physiological functions and operational mechanism of its closest homolog, the TRPV2 channel. TRPV2 from rats and mice can be activated by extreme temperature above 52°C (*Caterina et al., 1999*; *Neeper et al., 2007*; *Qin, 2011*; *Yao et al., 2011*), however, TRPV2 knock-out mice have normal sensitivity to noxious heat (*Park et al., 2011*) and the TRPV2 channel from humans is insensitive to heat (*Neeper et al., 2007*; *Qin, 2011*; *Yao et al., 2011*). Thus, temperature sensing does not appear to be a conserved function of the TRPV2 channel (*Peralvarez-Marin et al., 2013*). Indeed, the absence of high-affinity selective activators and inhibitors of TRPV2 has made exploration of this channel and its functions challenging (*Peralvarez-Marin et al., 2013*), although the channel can be activated with low affinity by non-selective agonists such as 2-aminoethoxydiphenyl borate (2-APB) (*Hu et al., 2004*; *Juvin et al., 2007*; *Neeper et al., 2007*), probenecid (*Bang et al., 2007*) and cannabinoids (*Qin et al., 2008*), and inhibited by ruthenium red. The TRPV2 channel is more widely expressed compared to TRPV1, and has been identified in sensory neurons, spinal cord, brain, pancreas, muscle and the immune system (*Peralvarez-Marin et al., 2013*). Although relatively few functional deficits have been observed in TRPV2 knock-out mice other than increased perinatal lethality (*Park et al., 2011*; *Peralvarez-Marin et al., 2013*), the channel has been proposed to play important roles in osmosensation, mechanosensation, innate immunity and cancer (*Katanosaka et al., 2014*; *Link et al., 2010*; *Monet et al., 2010*; *Peralvarez-Marin et al., 2013*; *Shibasaki et al., 2010*; *Zanou et al., 2015*).

The goal of the present study was to confirm the location of the binding site for vanilloids on TRPV1, and to explore whether TRPV2 channels share common gating mechanisms with TRPV1. The recent cryo-EM structure of TRPV2 reveals that the protein adopts a structure that is very similar to TRPV1 (*Zubcevic et al., 2016*), suggesting that the gating mechanisms of the two TRP channels may be more similar than their distinct pharmacology and functional properties might suggest. We began by investigating whether biochemically well-behaved domains of TRPV1 could be identified that are competent to bind vanilloids such as resiniferatoxin (RTx). Although we succeeded in producing

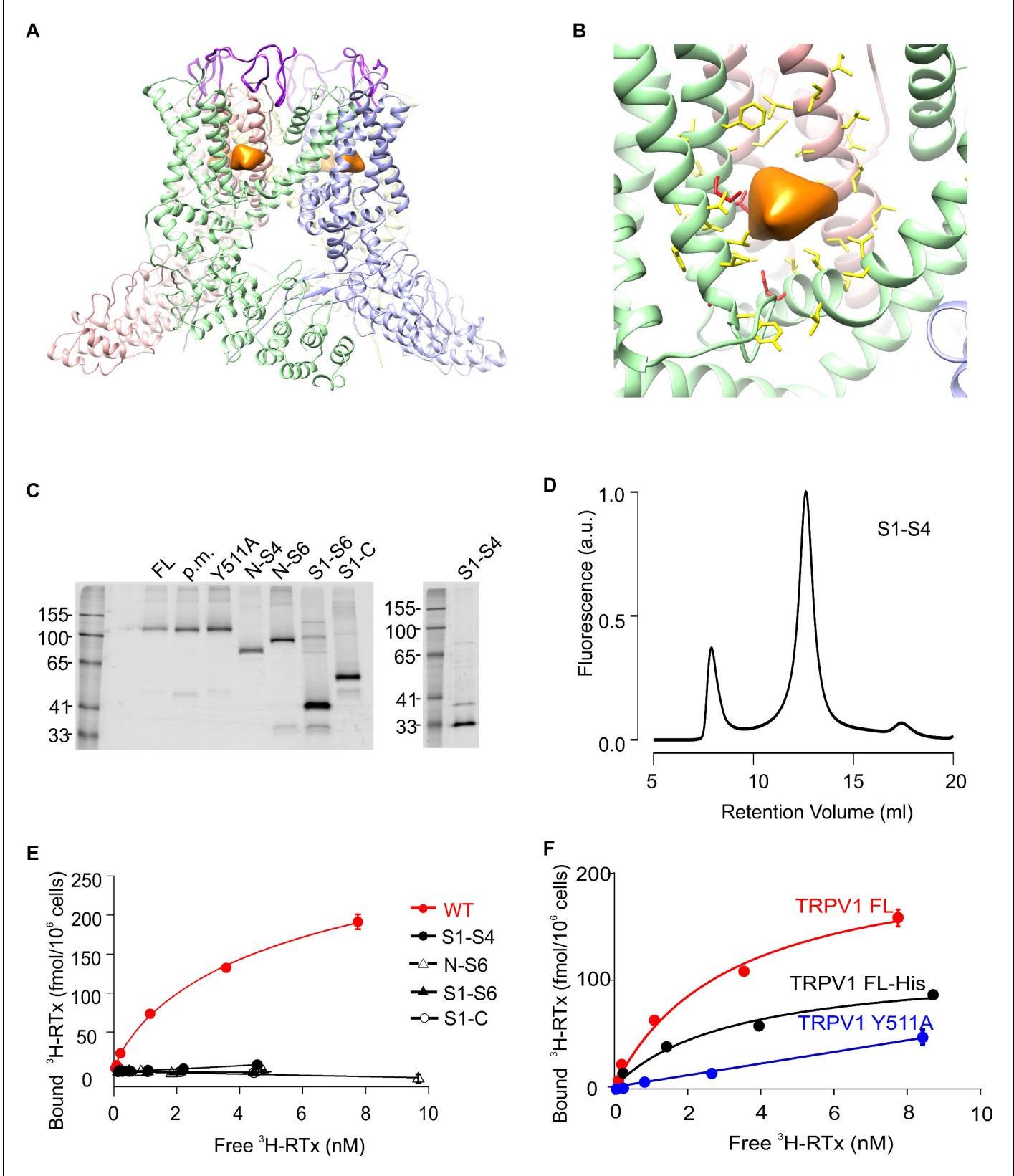

**Figure 1.** Structure of TRPV1 and characterization of TRPV1 constructs expressed in yeast. (**A**) Side view of the RTx/DkTx bound cryo-EM structures of TRPV1 refined using the NMR structure of DkTx (*Bae et al., 2016*). The EM density tentatively assigned to RTx is colored orange and the backbone of

*Figure 1 continued*

DkTx is colored bright purple. (**B**) Close-up view of the RTx EM density (orange) with residues closer than 12.5 Å from the center of mass of that density shown in stick representation. The four residues studied here (S512, T550, M547, E570) are colored red and the others yellow. (**C**) In-gel fluorescence of SDS-solubilized membranes from *S. cerevisiae* cells overexpressing GFP-tagged TRPV1 domain constructs using the following abbreviations: FL (Full-lenth), p.m. (pore mutant; deletion of 629–647)(*Garcia-Sanz et al., 2004*), N-S4 (N terminus through 575), N-S6 (N-terminus through 704), S1-S6 (423 to 704), S1-C (423 to C terminus) and S1-S4 (431 to 575). (**D**) Fluorescence-coupled size exclusion chromatography (FSEC) traces of GFP-tagged TRPV1 S1-S4 domain solubilized in DDM. The first peak is the void volume and the second corresponds to monomeric domain. (**E**) $^3$H-RTx binding to intact *S. cerevisiae* cells expressing full-length TRPV1 and its separate domains, all of which contain GFP tags on their C-termini. Data were normalized to number of cells because the trucated constructs express to higher levels compared to TRPV1, and data were corrected for non-specific binding as described in Materials and methods. Data points are the mean $\pm$ S.E.M. for triplicate determinations. Smooth function for full-length GFP-tagged TRPV1 is a fit of the Hill equation to the data with $K_d$ and Hill slope (nH) values of 7.0 nM and 0.78 for TRPV1. (**F**) Binding of $^3$H-RTx to *S. cerevisiae* cells containing GFP-tagged TRPV1, N-terminus His-tagged TRPV1 (without GFP), or dual GFP- and His-tagged TRPV1 construct harboring the Y511A mutation. Smooth functions are fits of the Hill equation to the data with $K_d$ and nH values of 7.2 nM and 0.82 for TRPV1, 30 nM and 0.6 for TRPV1-His, and >238 nM and 1.1 for TRPV1 Y511A. Data points are the mean $\pm$ S.E.M. for triplicate determinations.

truncated constructs of TRPV1, only the full-length channel binds RTx with high affinity. We also discovered that DkTx binding to the external pore of the full-length channel can enhance the affinity of RTx, providing critical support for an allosteric model for activation of TRPV1 (*Jara-Oseguera et al., 2016*). Using the cryo-EM structure of TRPV1 as a guide, we interrogated the putative vanilloid binding pocket with mutations in TRPV1 and our results confirm the location of the vanilloid binding site seen in the cryo-EM structure. We also attempted to engineer vanilloid sensitivity into the TRPV2 channel and made the remarkable discovery that only two mutations in TRPV2 are sufficient to enable vanilloids to bind with high affinity and to activate the channel. Single channel recordings of the vanilloid-sensitive TRPV2 channel show that high open probability ($P_o$) can be achieved with RTx, and that the permeation properties of TRPV2 also resemble those of TRPV1.

## Results

### Characterization of full-length and truncated constructs of TRPV1

Our initial objective was to see if we could define minimal domains of TRPV1 that were competent to bind the vanilloid RTx. Many of the previously identified mutants reported to alter RTx binding or capsaicin activation are localized within the S1-S4 domain of TRPV1 (*Elokely et al., 2016*; *Gavva et al., 2004*; *Jordt and Julius, 2002*; *Yang et al., 2015*), and in structurally related Kv channels, the S1-S4 region has been shown to form an autonomous domain (*Jiang et al., 2003a*; *2003b*) that is capable of imparting strong voltage-sensitivity when grafted onto relatively voltage-insensitive K$^+$ channels (*Ben-Abu et al., 2009*; *Lu et al., 2001*; *2002*). We began by expressing rTRPV1 and a series of truncated constructs in *Saccharomyces cerevisiae* using a previously described expression vector with a C-terminal GFP tag optimized for fluorescence-based screening of eukaryotic membrane protein expression and stability (*Drew et al., 2008*; *Parker and Newstead, 2014*). In addition to the full-length rTRPV1, we expressed the truncated constructs containing the N-terminus through S4 (N-S4), the N-terminus through S6 (N-S6), S1 through S6 (S1-S6) and S1 through the C-terminus (S1-C), as well as controls such as the Y511A mutant and a deletion within the outer pore (pore mutant) thought to stabilize the full-length rTRPV1 channel (*Garcia-Sanz et al., 2004*). Crude membrane preparations of small-scale *S. cerevisiae* trials of full-length TRPV1 and truncations yielded clean, prominent bands in Tris-Glycine gels imaged with in-gel fluorescence (*Figure 1C*), indicating robust expression of the proteins without significant degradation. Constructs containing the S1-S4 or S1-S6 domains of TRPV1 displayed good monodispersity when evaluated with fluorescence-coupled size exclusion chromatography (FSEC) using a range of detergents previously used to successfully solubilize eukaryotic membrane proteins (e.g. *Figure 1D*). Although purification attempts were initially promising for the S1-S4 domain, with clear protein bands visible in elution during the first step of purification, cleavage of GFP with TEV protease led to aggregation and precipitation (not shown).

To evaluate whether these constructs were competent to bind vanilloids, we initially measured the binding of [$^3$H]-RTx (*Szallasi and Blumberg, 1999*; *Szallasi et al., 1999*) to intact yeast cells

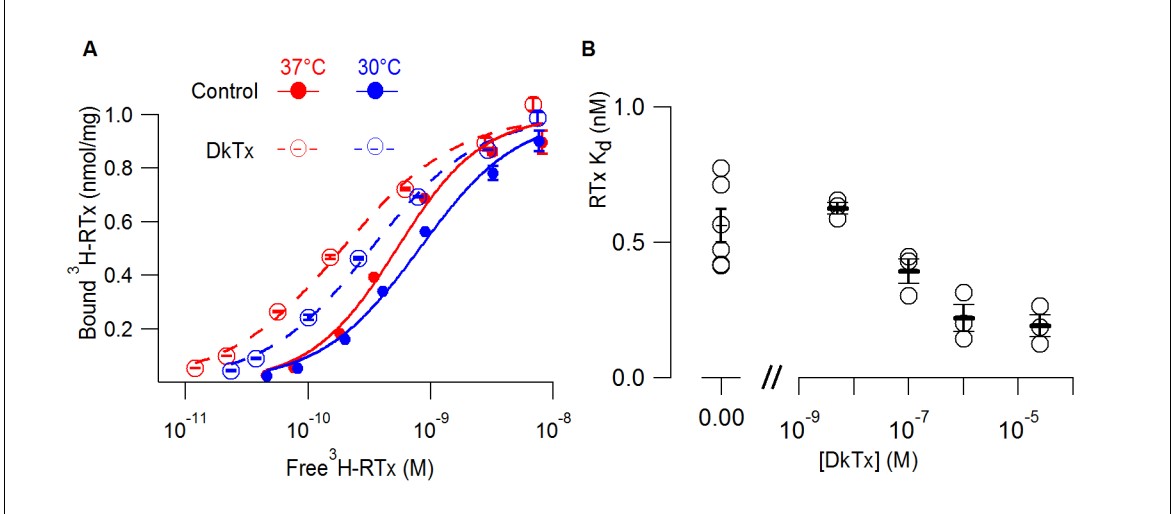

**Figure 2.** DkTx allosterically modulates the affinity of RTx for the TRPV1 channel. (**A**) $^3$H-RTx binding to *S. cerevisiae* membranes containing full-length 1D4-tagged TRPV1 in the presence or absence of DkTx (0.7 μM), normalized to total protein per sample. Smooth functions are fits of the Hill equation to the data with $K_d$ and Hill slope (nH) values 0.53 nM (nH=1.24) in control and 0.19 nM (nH=0.92) in the presence of DkTx at 37°C. At 30°C $K_d$ and nH values were 0.8 nM (nH=1.1) in control and 0.33 nM (nH=1) in the presence of DkTx. Data points are mean $\pm$ S.E.M. for triplicate determinations. (**B**) RTx $K_d$ values in the absence and presence of different concentrations of DkTx measured at 37°C. Values for individual cells are shown as circles and the mean $\pm$ S.E.M. as bars for between 3 to 5 separate experiments, each of which were determined in triplicate.

expressing each of the constructs. Although protein levels were comparable for these constructs based on GFP fluorescence, only the full-length TRPV1 channel displayed saturable $^3$H-RTx binding to intact cells, with a $K_d$ of 7.0 nM (*Figure 1E*). To demonstrate that only the full-length GFP-tagged channel presents specific high-affinity binding to RTx, we studied the Y511A mutant that disrupts vanilloid binding (*Jordt and Julius, 2002*; *Yang et al., 2015*), and found that the mutant greatly diminished RTx binding to GFP-tagged TRPV1 (*Figure 1F*). We also measured the binding of RTx to a His-tagged TRPV1 construct and observed that RTx bound with an affinity that was comparable to that observed with GFP-tagged TRPV1 (*Figure 1F*).

## Allosteric interactions between DkTx and RTx

The polymodal activation of TRPV1 has been widely conceptualized using allosteric models wherein binding of activators (such as vanilloids, protons, DkTx or voltage) are allosterically coupled to opening of the pore (*Brauchi et al., 2004*; *Jara-Oseguera et al., 2016*; *Matta and Ahern, 2007*). The allosteric nature of these interactions has begun to be explored using electrophysiological measurements of channel function (*Jara-Oseguera et al., 2016*; *Yao et al., 2010*), but has thus far not been investigated biochemically. We therefore tested whether binding of DkTx can enhance the affinity of TRPV1 for RTx, which would be predicted by an allosteric model wherein the two activators promote opening by binding more tightly to the open state. For these experiments we used a 1D4-tagged TRPV1 construct where expression in *S. cerevisiae* was driven by a constitutive promoter and we performed binding assays on yeast membranes where the vanilloid would have free access to the channel protein (see Methods). Under these conditions, we measured a $K_d$ for RTx binding to 1D4-tagged TRPV1 of 0.6 nM under control conditions, about ten fold higher affinity than what we measured in intact cells using GFP-tagged TRPV1. When we measured the concentration-dependence of RTx binding to 1D4-tagged TRPV1 in the presence of DkTx, we observed that the tarantula toxin enhanced the affinity for RTx by about 3-fold (*Figure 2A*). The effects of DkTx were concentration-dependent (*Figure 2B*) and were observed independent of whether the binding assay was performed at 37°C (our standard temperature) or at 30°C (*Figure 2A*). These results demonstrate that DkTx binds to our preparation of TRPV1, and that binding of DkTx and RTx are allosterically coupled.

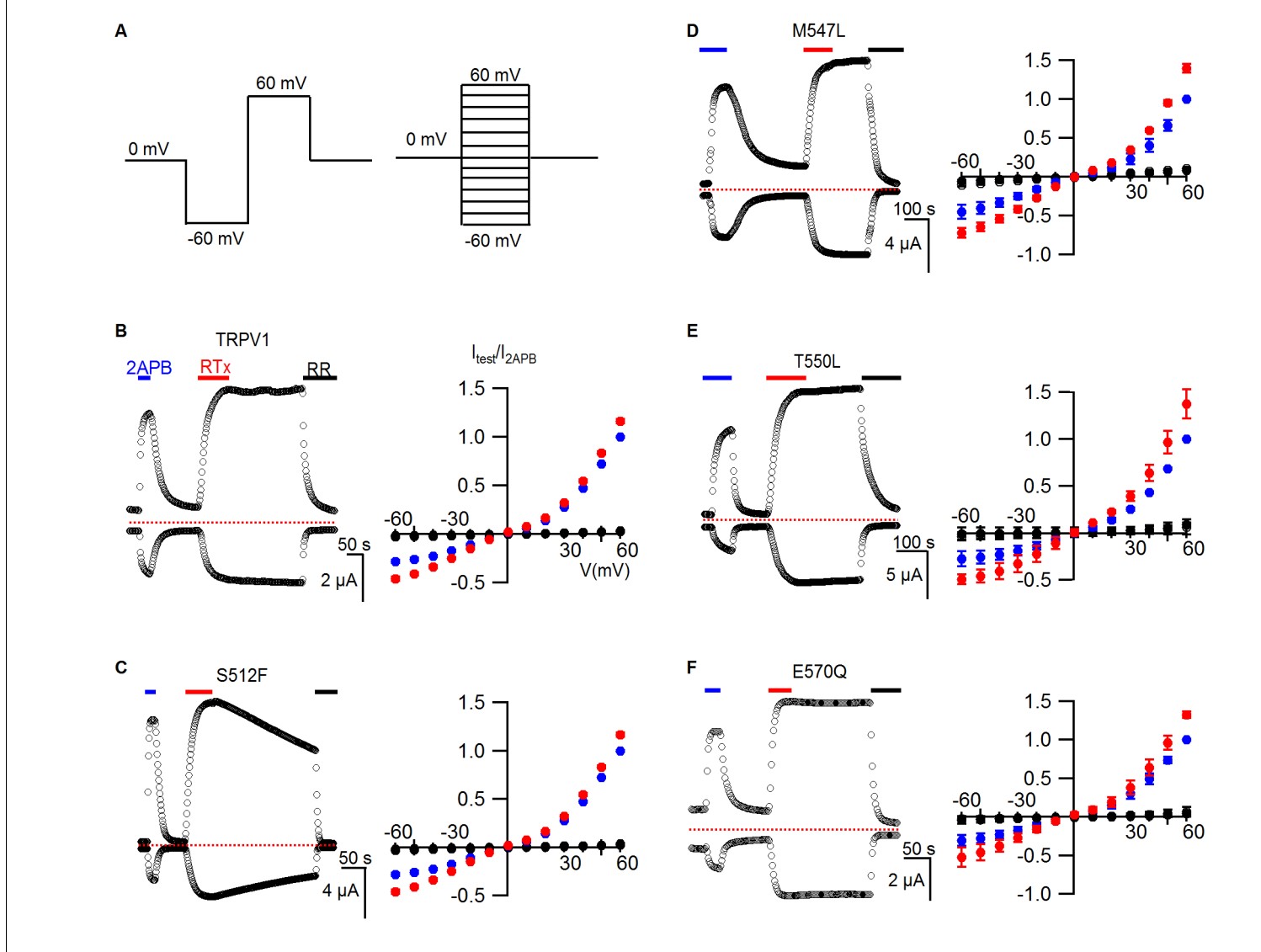

**Figure 3.** RTx sensitivity of individual mutations in TRPV1. (A) Voltage protocols used to measure time courses for activation of TRPV1 (left) or I-V relations (right). (B, left panel) Representative time course for WT TRPV1 activation in response to 2-APB and RTx measured from outward currents at +60 mV and from inward currents at -60 mV. Pulses were given every 3 s. The colored horizontal lines indicate application of agonists (2 mM 2-APB, 100 nM RTx, 50 μM RR). The dotted horizontal line indicates the zero-current level. (B, right panel) Mean normalized I-V relations obtained in control (open symbols, not visible), 2-APB (blue), RTx (red) and RR immediately after RTx (black filled symbols). Currents were normalized to the value in the presence of saturating 2 mM 2-APB. Data are expressed as mean ± S.E.M. (n = 4). (C-F) Time courses and I-V relations (n=3–5) for individual mutations as indicated, studied using the same protocols as in (A,B).

The following figure supplement is available for figure 3:

**Figure supplement 1.** Sequences of the transmembrane regions of rat TRPV1-4 channels.

## Interrogating the vanilloid binding pocket in TRPV1

To further investigate the putative vanilloid binding pocket in TRPV1, we inspected the cryo-EM structure of TRPV1, looking for residues that are located nearby to the putative RTx density (*Figure 1B*; regions/residues highlighted in yellow) and that differ between TRPV1 and TRPV2 channels (*Figure 3—figure supplement 1*; regions highlighted in magenta). This analysis identified S512 (F472 in TRPV2), M547 (L507 in TRPV2), T550 (L510 in TRPV2) and E570 (Q530 in TRPV2) as

potentially important determinants of RTx binding to TRPV1, consistent with previous experimental demonstrations that mutations of these residues weaken activation of TRPV1 by vanilloids (*Elokely et al., 2016*; *Gavva et al., 2004*; *Jordt and Julius, 2002*; *Yang et al., 2015*). To quantify the contribution of each of these residues to TRPV1 channel activation by RTx under our experimental conditions, we mutated each to the corresponding residue in TRPV2. We then expressed each construct in oocytes and used the two electrode voltage-clamp technique to study activation by RTx and by the non-specific TRPV channel activator 2-APB. We recorded time courses for channel activation by each agonist while stepping the membrane potential from −60 to +60 mV every 3 s from a holding potential of 0 mV using an external recording solution containing 100 mM KCl (internal $K^+$ concentration in a healthy oocyte is 100 mM) (*Figure 3A*, left). We also recorded current-voltage (I-V) relations by stepping over a wider range of voltages in the absence and presence of activators, or in the presence of ruthenium red (RR), a non-specific inhibitor of TRP channels (*Figure 3A*, right). Because RTx has high affinity for TRPV1 and its interaction with perfusion tubing makes it difficult to reliably control its concentration in the low nM range, we applied the vanilloid at 100 nM, a concentration that is saturating based on our RTx binding assay (*Figure 2A*). In the case of the wild-type TRPV1 channel, 2-APB produced reversible activation of outward and inward currents, RTx produced effectively irreversible activation on the time-scale of our recordings, and RR blocked the RTx-activated currents (*Figure 3A*, left). The I-V relations we recorded gave results that were consistent with the time courses, and we used these to present results from a population of cells (*Figure 3A*, right). Similar results were obtained with each of the four TRPV1 mutants (*Figure 3B–E*), with the exception being the S512F mutant, where we could readily see dissociation of RTx after removing the vanilloid from the recording chamber (*Figure 3B*), suggesting that this mutant weakens the affinity of RTx. Although previous studies suggest that some of these residues are determinants of vanilloid-sensitivity in TRPV1 (*Gavva et al., 2004*; *Jordt and Julius, 2002*; *Yang et al., 2015*), their individual substitution by the equivalent residues in TRPV2 is not sufficient to produce large disruptions in the activation by a saturating concentration of RTx.

We next combined all four mutants to make a quadruple mutant (TRPV1 QM), which completely lost sensitivity to RTx while retaining sensitivity to 2-APB (*Figure 4A*), suggesting that differences between TRPV1 and TRPV2 at some or all of these positions can explain the differential sensitivity of the two TRP channels to RTx. Although we did not measure the binding affinity of RTx for the TRPV1 QM, leaving open the possibility that the vanilloid binds without activating the channel, this is unlikely because individually the T550I and M547L mutants dramatically weaken RTx binding affinity (*Gavva et al., 2004*). To further define which residues are most critical, we made double mutants where we combined S512F, the mutation that speeds RTx dissociation, with each of the other three mutations. Both S512F/M547L and S512F/T550L channels lost sensitivity to RTx, whereas the S512F/E570Q mutant retained RTx sensitivity (*Figure 4B–D*), indicating that S512, M547 and T550 are more important determinants of TRPV1 activation by RTx than E570. We also examined all four possible triple mutants, and only the mutant that left S512 intact retained sensitivity to RTx (*Figure 4—figure supplement 1*). Collectively, these results suggest that 1) S512 is the most critical determinant of RTx sensitivity, 2) M547 and T550 are also important, and 3) E570 may or may not contribute to the differential sensitivity of TRPV1 and TRPV2 to RTx.

## Engineering vanilloid sensitivity into TRPV2

We began investigating the properties of TRPV2 by testing whether the channel is sensitive to RTx. Although TRPV2 could be activated by 2-APB, the channel was completely insensitive to RTx (*Figure 5A*), consistent with previous reports that this TRP channel is insensitive to capsaicin (*Caterina et al., 1999*). We then introduced all four TRPV1 residues that contribute to RTx sensitivity into TRPV2 to make the TRPV2 QM, and observed that the channel remained insensitive to capsaicin, but was robustly activated by the higher affinity vanilloid RTx (*Figure 5B*). RTx produced much stronger activation when compared to 2-APB, and we observed little evidence for dissociation of the vanilloid after its removal from the recording chamber (*Figure 5B*), suggestive of strong and stable binding to TRPV2 QM. To examine the strength of RTx binding to the TRPV2 channel, we undertook binding assays after expressing 1D4-tagged TRPV2 and TRPV2 QM in *S. cerevisiae* and isolating yeast membranes, similar to the experiments described above with TRPV1 (*Figure 2*). Although we only observed weak binding of RTx to TRPV2, the TRPV2 QM displayed robust and high-affinity RTx binding with a $K_d$ of 18 nM (*Figure 5C*). The affinity of RTx for the TRPV2 QM is about 30-fold lower

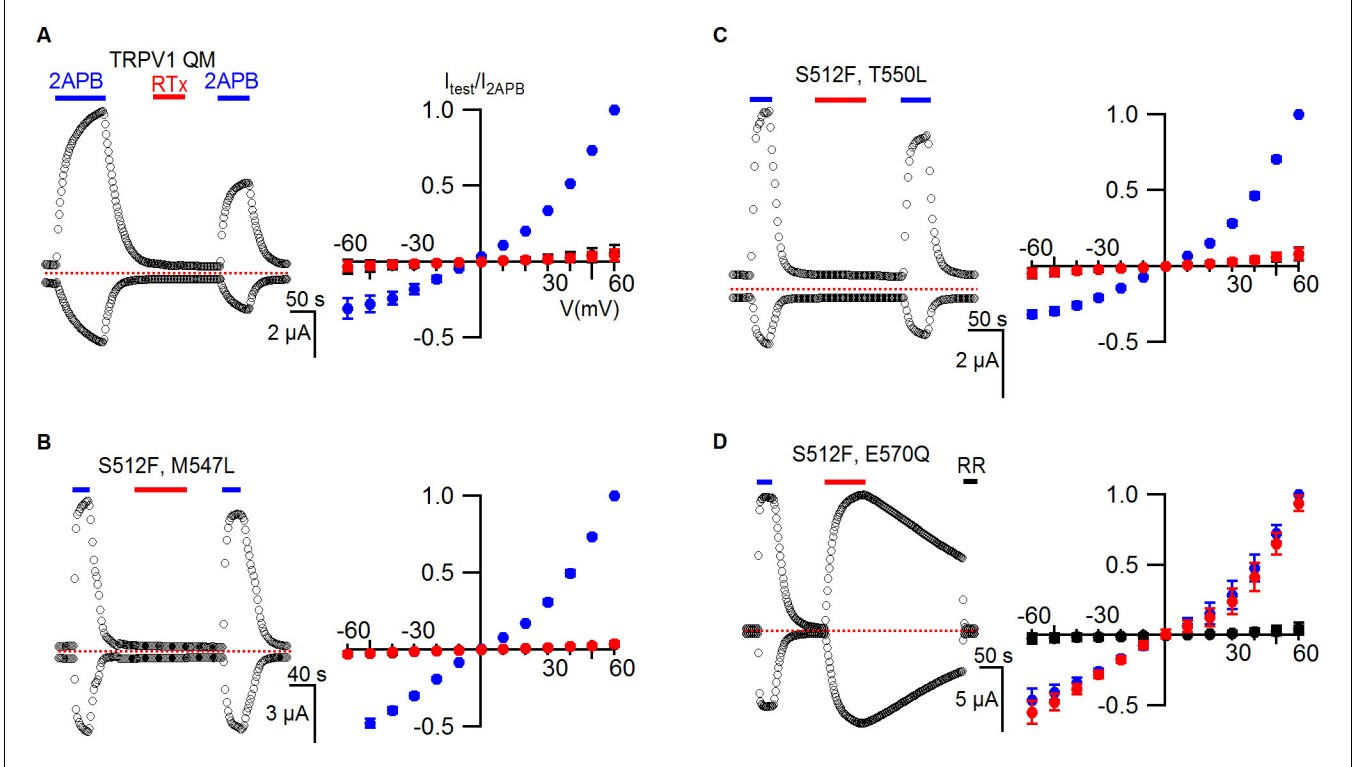

**Figure 4.** TRPV1 quadruple mutant and double mutant responses to RTx. (**A**, left panel) Representative time courses for TRPV1 QM (S512F, M547L, T550L, E570Q) activated by 2-APB (2 mM) and RTx (100 nM) measured at +60 and -60 mV. RR (50 μM) was applied after RTx. Same pulse protocol as *Figure 3*. The dotted horizontal line indicates the zero-current level. (**A**, right panel) Mean normalized I-V relations obtained in control (open circles; not visible), 2-APB (blue) and RTx (red). Currents were normalized to the value in the presence of 2 mM 2-APB . Data are expressed as mean ± S.E.M. (n = 3). (**B-D** Left and right panel) Time course and I-V relations of double mutants of TRPV1 activated by 2-APB (2 mM) or RTx (100 nM) obtained the same way as in (**A**). For the I-V relations in (**D**), currents measured when RR was applied immediately after RTx are shown as filled black symbols. Currents were normalized to the value in the presence of 2 mM 2-APB (n=3–5).

The following figure supplement is available for figure 4:

**Figure supplement 1.** TRPV1 triple mutant responses to RTx.

compared to TRPV1, suggesting that other differences between TRPV1 and TRPV2 may influence RTx affinity. Indeed, there are eight other residues located near the RTx binding pocket that differ between TRPV1 and TRPV2 (see *Figure 3—figure supplement 1*), which may also explain why the TRPV2 QM is not activated by capsaicin. To examine the relative importance of the four residues for mediating RTx sensitivity in the TRPV2 QM, we constructed triple mutants and evaluated their sensitivity to RTx electrophysiologically. All three triple mutants containing the F472S mutation (equivalent to S512 in TRPV1) were sensitive to RTx, whereas the one triple mutant that retained F472 was insensitive to RTx (*Figure 5D*), consistent with a dominant role of S512 that we observed in TRPV1 (*Figure 4*; *Figure 4—figure supplement 1*). We also investigated the RTx sensitivity of three double mutants containing the F472S mutation, and observed that either L507M or L510T rendered the TRPV2 channel sensitive to RTx (*Figure 5—figure supplement 1*), again matching our results for loss of function experiments in TRPV1 and demonstrating that only two point mutations are required to make the TRPV2 channel sensitive to RTx. Notably, the F472S single mutation is likely not sufficient to confer RTx sensitivity to TRPV2, as the F472S/Q530E was unresponsive to the vanilloid (*Figure 5—figure supplement 1C*).

Previous studies on TRPV1 have shown that vanilloids such as capsaicin promote the open state with high efficacy, with maximal open probabilities ($P_o$) of around 0.8 to 0.9 at saturating concentrations of the agonist (*Hui et al., 2003*; *Oseguera et al., 2007*; *Premkumar et al., 2002*). To explore

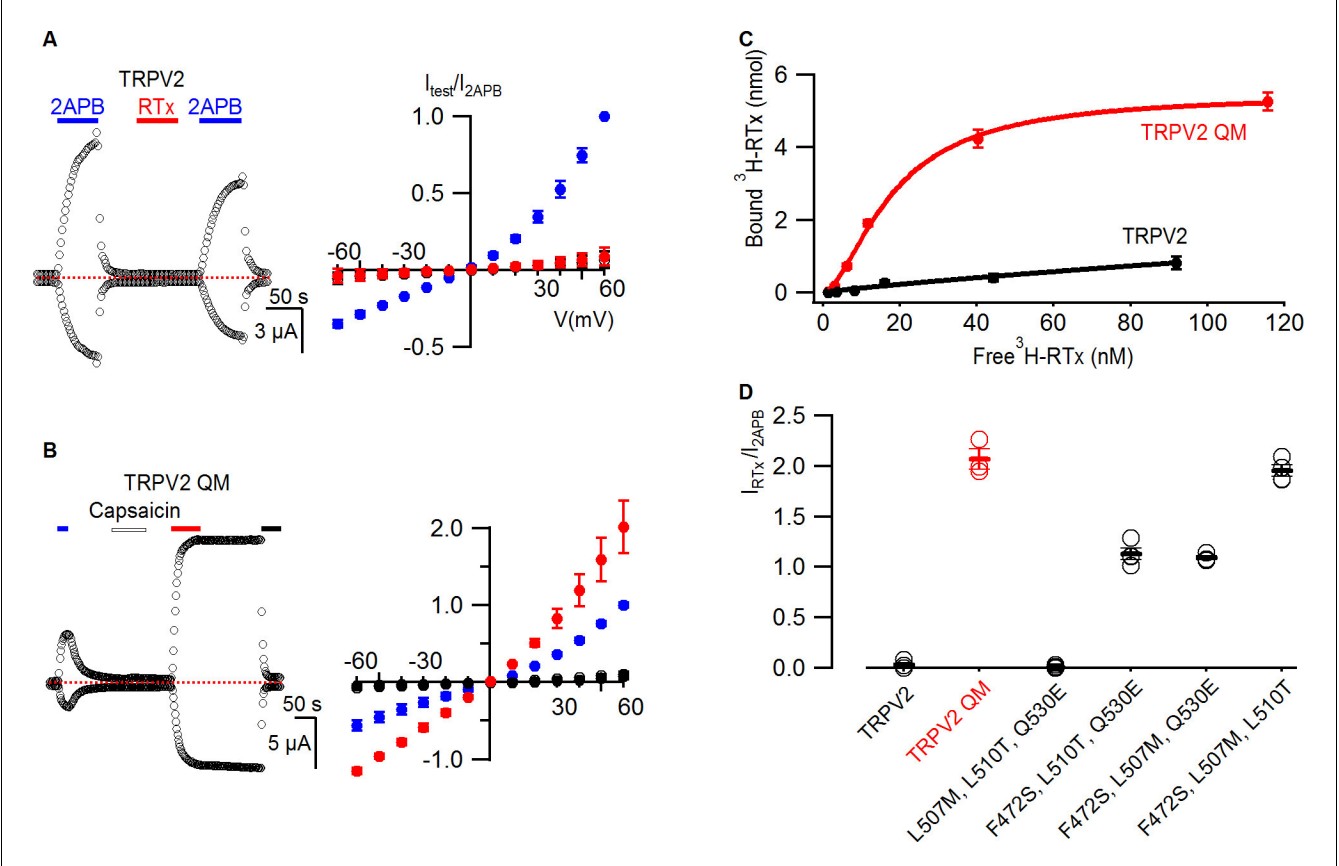

**Figure 5.** RTx binds to and activates the TRPV2 QM. (**A**, left panel) Representative time courses for WT TRPV2 in the presence of 2-APB (4 mM) or RTx (100 nM) measured at +60 and -60 mV. Same pulse protocol as **Figure 3**. The dotted horizontal line indicates the zero-current level. (**A**, right panel) Mean normalized I-V relations under different conditions using the color codes from **Figure 3**. Currents were normalized to the value in the presence of 2-APB (4 mM). Data are expressed as mean ± S.E.M. (n = 3). (**B**, left panel) Response of the quadruple mutant of TRPV2 (F472S, L507M, S510T, Q530E) to 2-APB (4 mM),capsaicin (50 μM) and RTx (100 nM) obtained as in (**A**). (**B**, right panel) Mean normalized I-V relations under different conditions using the color codes from **Figure 3**. Data are expressed as mean ± S.E.M. (n=4–8). (**C**) Binding of $^3$H-RTx to membranes containing 1D4-tagged TRPV2 QM (red) and 1D4-tagged WT TRPV2 (black). Membranes were prepared from yeast expressing the two constructs to comparable levels as judged by westerns using a 1D4 mAb and equal amounts of membranes were used for assessing binding of RTx to the two TRPV2 constructs, and data were corrected for non-specific binding as described in Materials and methods. Smooth functions are fits of the Hill equation to the data with $K_d$ and nH values of 18 nM and 1.7 for TRPV2 QM and of > 200 nM and 0.9 for wt TRPV2. (**D**) Summary of the response of TRPV2, TRPV2 QM and thee triple mutants to RTx. Currents measured in response to RTx (100 nM) at +60 mV are normalized to that measured in response to 2-APB (4 mM). Values for individual cells are shown as circles and the mean ± S.E.M. as bars.

The following figure supplement is available for figure 5:

**Figure supplement 1.** Responses of TRPV2 double mutants to RTx.

the extent to which RTx promotes the open state of the TRPV2 QM, we investigated the unitary properties of the channel using inside-out patch recordings obtained from transiently transfected HEK293 cells. In a patch containing a single TRPV2-QM channel, we observed no spontaneous channel activity prior to application of any agonist (data not shown), but a few seconds of exposure to a solution containing 100 nM RTx resulted in robust channel activation (**Figure 6A**). The channel entered a mode where the $P_o$ was very close to 1 for the entirety of the recording (approximately 8 min), presenting only brief closures (**Figure 6A and E**). The conductance of the main conducting state was 101 pS in symmetrical 140 mM NaCl at +90 mV (**Figure 6A, D and G**), but occasional sub-conducting states could also be detected (**Figure 6C**). We observed pronounced rectification to the single channel conductance, with a value of 28 pS at -90 mV (**Figure 6B, D and G**). These results reveal that the single channel conductance of TRPV2 QM is quite high and that it rectifies, two

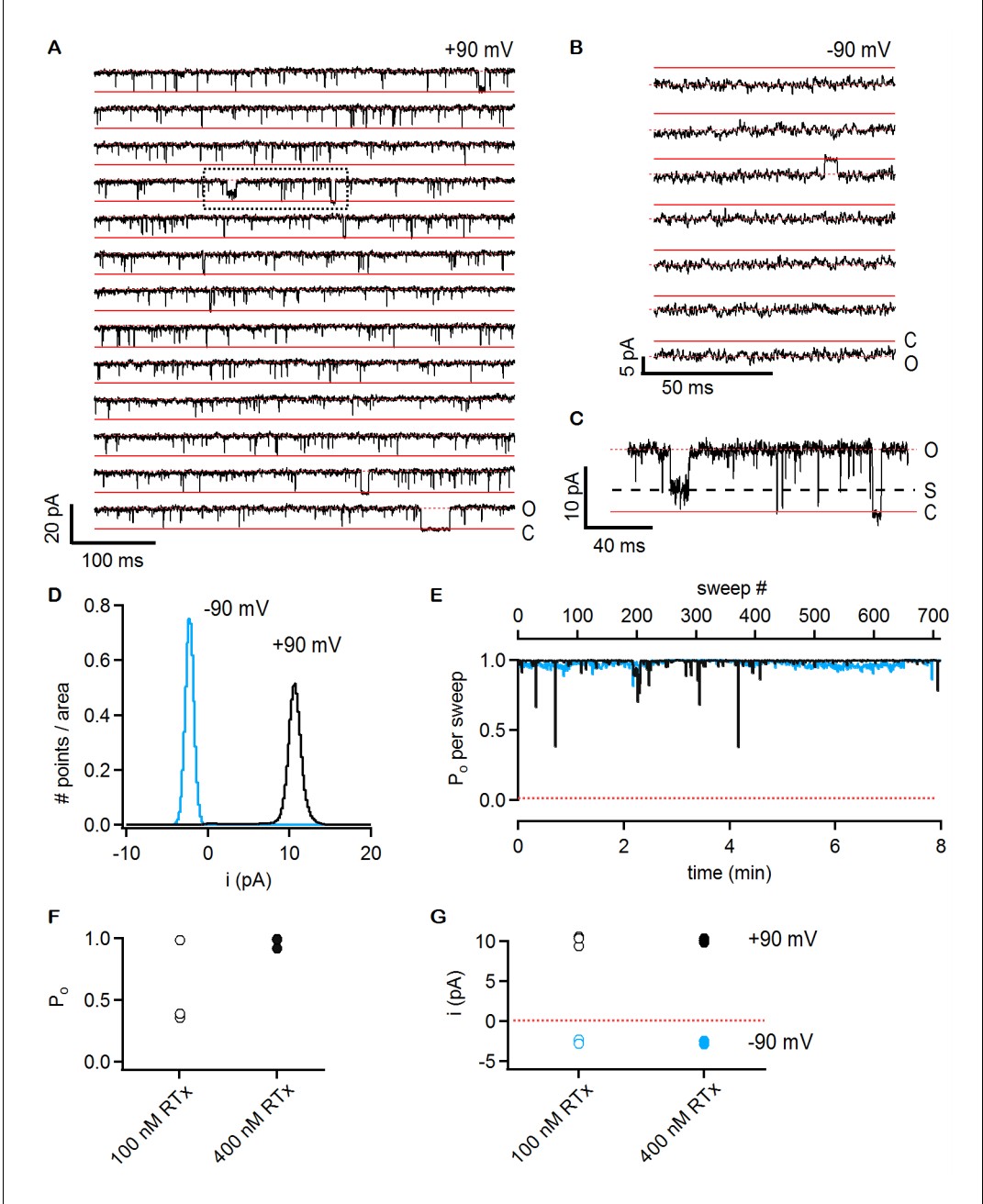

**Figure 6.** RTx activates the TRPV2 QM channel with high efficacy and affinity. (A) Single-channel recordings of TRPV2 QM obtained from an inside-out patch at +90 mV in the presence of 100 nM RTx. Currents were elicited by 500 ms pulses to +90 mV from a holding potential of −90 mV applied every 200 ms. A trace without openings obtained in control solution prior to RTx application was used for leak-subtraction. The red continuous horizontal lines indicate the zero-current level (closed channel - C) and the dotted lines the current level for an open channel - O. The blue dotted rectangle denotes a portion of the recording that is shown in higher magnification in (C). (B) Single-channel recordings from the patch in (A) obtained at the holding potential of -90 mV. The continuous red lines are the zero current level (closed channel – C) and the dotted lines denote an open channel (O). (C) Magnification of the single-channel recordings in the dotted black box in (A) showing the occurrence of subconductance levels, denoted by (S) and the dashed black line. (D) Normalized all-points histograms from all recordings obtained at +90 (black) and -90 mV (blue) from the patch in (A-C). Histograms were normalized to the total area under the curve. (E) $P_o$ per sweep as a function of recording time calculated for all recordings at +90 (black) and -90 mV (blue) from the same patch in (A-D). The dotted red line denotes a $P_o$ of 0. (F) Mean $P_o$ values for TRPV2 QM at 100 nM RTx (open symbols) and +90 mV calculated from the single-channel patch in (A-E) and

*Figure 6 continued on next page*

*Figure 6 continued*

from two other patches with two channels each (n = 3 independent patches), together with $P_o$ values at 400 nM RTx (closed symbols) calculated from two single-channel patches and two patches containing two channels each (n = 4 independent patches). (**G**) Single-channel current amplitudes for the full-conductance level of an open TRPV2 QM channel in the presence of 100 nM (open symbols) or 400 nM (closed symbols) RTx at +90 (black) or -90 mV (blue) obtained from the same patches as the $P_o$ values in (**F**). The dotted red line denotes the zero-current level.

The following figure supplement is available for figure 6:

**Figure supplement 1.** Single-channel activity of TRPV2 QM in the presence of 100 nM and 400 nM RTx.

properties that are similar to what has been observed for TRPV1 (*Hui et al., 2003*; *Oseguera et al., 2007*; *Premkumar et al., 2002*). Two other patches containing two TRPV2 QM channels each presented a slightly lower $P_o$ at 100 nM RTx (*Figure 6F*), as the channels in these patches appeared to enter long-lived resting states between bursts of maximal $P_o$ (*Figure 6—figure supplement 1A–C*). When a higher concentration of RTx was used (400 nM), long-lived closed states could be detected in two patches with a single channel and two patches with two channels each (*Figure 6F* and *Figure 6—figure supplement 1D–F*). From these results we conclude that the permeation properties of TRPV2 channels are similar to TRPV1 channels and that RTx promotes the open state with high efficacy.

## Discussion

The goal of the present study was to further interrogate the putative vanilloid binding pocket identified in the recent cryo-EM structures of the TRPV1 channel (*Cao et al., 2013*) and to investigate the extent to which the enigmatic TRPV2 channel shares functional mechanisms in common with TRPV1 (*Peralvarez-Marin et al., 2013*). Our biochemical results with expressed TRPV1 constructs and the RTx binding assay demonstrate that the full-length protein expressed in yeast is competent to bind RTx with high affinity, but that the S1-S4 and S1-S6 domains are not sufficient to observe convincing binding of the vanilloid (*Figure 1E,F*). The RTx binding site identified in the cryo-EM structure of TRPV1 is located at the interface between the S1-S4 domain of one subunit and the S5-S6 domain of the adjacent subunit (*Cao et al., 2013*), and therefore one explanation would be that our truncated constructs do not bind RTx because they are possibly monomeric. We took advantage of the ability of the full-length construct to bind RTx to investigate whether we could observe allosteric coupling between the binding of the vanilloid and that of DkTx, a tarantula toxin that binds to the outer pore of TRPV1. We observed readily detectable shifts in the $K_d$ for RTx binding in the presence of DkTx (*Figure 2*), indicating that both toxins are able to bind to our preparation of TRPV1 and that they are allosterically coupled. Although this type of allosteric coupling is embodied in the allosteric models that have been widely used to understand the complex polymodal gating of TRPV1 seen in electrophysiological studies (*Jara-Oseguera et al., 2016*; *Latorre et al., 2007*), this is the first direct biochemical demonstration of allosteric coupling between two ligands that activate TRPV1.

In addition to providing further validation of the RTx binding site inferred from the cryo-EM structure of TRPV1 (*Figure 4*; *Figure 4—figure supplement 1*), our mutagenesis results in TRPV2 provide a new perspective with which to view this poorly understood TRP channel. Our results demonstrate that as few as two TRPV1 residues need to be introduced into TRPV2 in order for RTx to bind to the channel with high affinity and to fully activate the channel (*Figure 5*; *Figure 5—figure supplement 1*; *Figure 6*). This finding suggests that a quiescent RTx binding pocket already exists in TRPV2, requiring only subtle engineering to make it competent to bind RTx, and demonstrating that the same basic vanilloid-activation mechanism exists in TRPV2 channels. Our single channel recordings of TRPV2 show that RTx binding can achieve high $P_o$, and that the conductance of the open state is high and exhibits rectification (*Figure 6*), properties that are very similar to the TRPV1 channel (*Hui et al., 2003*; *Liu et al., 2009*; *Oseguera et al., 2007*; *Premkumar et al., 2002*). Collectively, these findings fit very well with the recent TRPV2 cryo-EM structure (*Zubcevic et al., 2016*), which shows that the channel adopts a structure that is remarkably similar to TRPV1 (*Liao et al., 2013*) (*Figure 7*). Our results are also compatible with a computational model of RTx docked into the

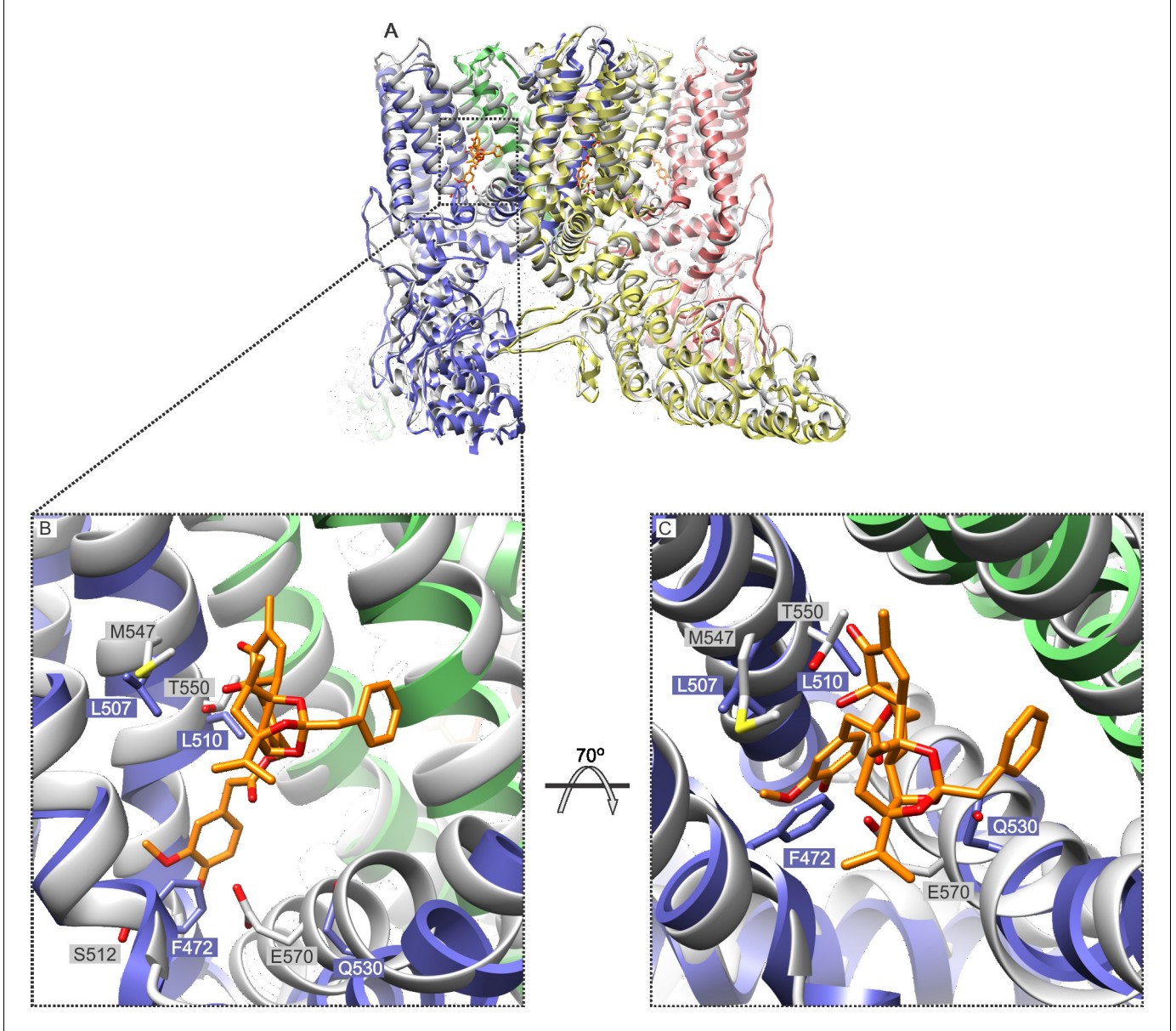

**Figure 7.** Comparison of the structures of TRPV1 and TRPV2 and the RTx binding pocket. (**A**) Side view of the superposition of the cryo-EM structure of the RTx/DkTx-bound rTRPV1 channel (gray) (*Cao et al., 2013*) with that of rabbit TRPV2 (colored by chain) (*Zubcevic et al., 2016*). RTx is shown in orange as docked by (*Elokely et al., 2016*). (**B,C**) Enlarged views of the RTx binding pocket showing the side chains in stick representation for the four mutated residues in TRPV2 QM numbered according to the rat sequence of the two proteins. TRPV2 subunits are colored by chain and TRPV1 is shown in gray.

structure of TRPV1, where the methoxyl and hydroxyl bearing aromatic ring of RTx hydrogen bond with S512 in TRPV1 (*Elokely et al., 2016*). If we superimpose the cryo-EM structures of TRPV1 and TRPV2, the native F472 in TRPV2 (corresponding to S512) would likely sterically clash with the methoxyl and hydroxyl groups of RTx, and could not support the stabilizing hydrogen bonds with the hydroxyl side chain of S512 (*Figure 7*), explaining why this position is so critical for vanilloid binding to both TRPV1 and TRPV2 channels.

The mutants we describe should be useful for obtaining structures of the open state of the TRPV2 channel in the presence of RTx, and for further exploring the gating mechanisms that differ between TRPV1 and TRPV2 channels. Proton activation, as well as regulation by external Na$^+$ (*Jara-Oseguera et al., 2016*), for example, appear to be unique features of TRPV1 channels. In

demonstrating that TRPV1 and TRPV2 share vanilloid-activation mechanisms, our results suggest that the TRPV2 channel should be an excellent tool for defining the sites and mechanisms of activation for stimuli that are specific to TRPV1. In addition, the residues mutated in the TRPV2 quadruple mutant are for the most part conserved among TRPV2, TRPV3, and TRPV4 channels (*Figure 3—figure supplement 1*), suggesting that despite differing functional properties and physiological roles, it is possible that the vanilloid gating pathway of TRPV1 and TRPV2 is also conserved in TRPV3 and TRPV4 channels.

## Materials and methods

### Expression and characterization of GFP-tagged TRPV1 constructs in *Saccharomyces cerevisiae*

Following a previously described *S. cerevisiae* membrane protein expression protocol (*Drew et al., 2008*; *Parker and Newstead, 2014*), small-scale expression trials were performed in 10 mL of -Leu selective medium with 2% glucose incubated at 30°C in an orbital shaker for ~22 hrs. The overnight culture was then diluted into 10 mL of Leu-containing media with 2% lactate to an $OD_{600}$ of 0.3, cells were grown to an $OD_{600}$ of 1.0 (12–20 hr), at which point protein expression was induced by adding 2% (wt/vol) galactose. Cells were harvested after 24 hr by centrifugation at 3000 g for 5 min. In-gel fluorescence was also performed on crude membrane preps of small-scale trials using glass beads and vortexing to break cells. To pellet crude membranes, broken cells were centrifuged at 20,000g at 4°C for 1 hr and the membrane fraction combined with (2x) Loading Buffer and loaded onto a Tris-Glycine (10%, 12%, 15% acrylamide depending on desired protein resolution) gel along with 2 µL of Invitrogen's Fluorescent BenchMark Ladder. Gels were imaged using a FLA-3000 fluorescent gel imager from FujiFilm.

Constructs that showed high levels of overexpression as assayed by whole cell fluorescence and low levels of degradation measured using in-gel fluorescence were selected for medium scale preparations (2L) to assess detergent solubility and quality of the protein upon solubilization. A 10 ml overnight culture of the –Leu selective medium containing 2% glucose was diluted into 2 L of Leu-containing media with 2% lactate to an $OD_{600}$ of 0.3, and the cells were then incubated to an $OD_{600}$ of 1.0 (for 12–20 hr), at which point protein expression was induced by adding 2% galactose. *S. cerevisiae* cells were broken in four passes at 25kpsi, 30kpsi, 35kpsi, and 40kpsi using an Avestin homogenizer. Debris was then removed by centrifugation at 15,000g for 10 min. Fluorescence was used to check cell lysis efficiency by measuring the fluorescence before and after spinning down debris. Membranes were then centrifuged from the supernatant at 41,000 RPM and 4°C for 2 hr in an OptimaTML-100K ultracentrifuge using a Ti45 rotor. 900 µL of membranes were transferred to a clean 1.5 mL Eppendorf tube, and 100 µL of 10% detergent (DDM) was added to obtain a 1% final detergent concentration. This membrane detergent mixture was then solubilized for 1 hr at 4°C with gentle agitation. Solubilized membranes were then centrifuged out at 100,000g for 45 min at 4°C in a desktop ultracentrifuge. The solubilized membranes were then transferred to a clean 1.5 mL tube before 500 µL were loaded onto the GE Superose 6 10/300 column coupled to a fluorometer.

Once it was determined that a construct showed both promising overexpression levels and indications of monodispersity upon detergent solubilization, large-scale cell cultures were prepared for purification. After membranes were prepared they were detergent solubilized in a beaker with stirring for 1 hr at 4°C. Unsolubilized membranes were removed by centrifugation for 1 hr at 4°C at 41,000 RPM in a Beckman Ultracentrifuge with the Ti45 rotor. Solubilization efficiency was calculated by measuring fluorescence before and after centrifugation. Solubilized membranes were then incubated with Ni-NTA resin for 2 hr with stirring at 4°C, the mixture poured through a column, and binding efficiency calculated by measuring the fluorescence of this solution with resin and flow through. The protein was then eluted in 250 mM imidazole, and cleaved during overnight dialysis at 4°C with tobacco etch virus (TEV) protease (2 mg TEV/1 mg protein as estimated by fluorescence) in 3,500 MWCO dialysis tubing with stirring in 1.5 L dialysis buffer (150 mM NaCl, 20 mM Tris, 3x CMC detergent). In a second step of purification, after dialysis, the presumably cleaved protein solution was filtered with a 22 µM filter, and passed through a Ni Sepharose High Performance HisTrap column. The flow-through, now devoid of His- and GFP tags, as well as His-tagged TEV, was then concentrated to 500 µL using a 50,000 MWCO Vivaspin 20 concentrator. Generally, for the S1-S4 and

S1-S6 of rTRPV1, aggregates formed during the TEV cleavage step, and if the second step of purification was performed no cleaved protein was visible in coomassie-stained gels.

## Expression of 1D4-tagged TRPV1 and TRPV2 constructs in *Saccharomyces cerevisiae*

The full-length TRPV1 and TRPV2 channels with 1D4 affinity tags (TETSQVAPA) on their C-termini were cloned into the YEpHIS vector and expressed in the BJ5457 strain of *S. cerevisiae* using the strong constitutively active PMA1 promoter, as previously described (*Moiseenkova-Bell et al., 2008*). After 48 hr of expression at 30°C in YPD media, cells were harvested by centrifugation and re-suspended in PBS buffer, supplemented with complete protease inhibitor cocktail (Roche, Indianapolis, IN) and 4-amidinophenylmethanesulfonyl fluoride hydrochloride (APMSF) (Sigma-Aldrich, St. Louis, MO). Cells were broken by passing through an Avestin homogenizer at the pressure of 35 kpsi, cell lysate was cleared by centrifugation at 8,000 g for 15 min and membranes were isolated by ultracentrifugation at 190,000 g for 1 hr. Membrane pellets were re-suspended in PBS buffer supplemented with protease inhibitors and 200 mM sucrose, flash-frozen in liquid nitrogen and stored at -70°C until RTx binding measurements.

## Radioligand binding assays

*S. cerevisiae* cells overexpressing TRPV1 constructs to be used in ligand binding assays on intact cells (*Figure 1E,F*) were grown as for small scale expression trials. 24 hr after protein induction cells were spun down at 4000g, the supernatant was removed and the pellet resuspended in 1x PBS, normalizing $OD_{600}$ to 75 for each sample, and aliquoted for storage at -80°C. $^3$H-RTx binding assays on cells were performed at 37°C as previously described (*Feng et al., 2015*), but where non-specific binding of RTx was measured using cells expressing GFP alone. The binding of RTx to TRPV1 and TRPV2 in membranes (e.g. *Figure 2A,B* and *5C*) was performed at 37°C unless otherwise stated using membranes prepared from large scale protein preparations of *S. cerevisiae* containing rat TRPV1 and TRPV2 with a 1D4 tag expressed using a constitutive promoter as in *Moiseenkova-Bell et al., 2008*. Data for binding of RTx to membranes was corrected for non-specific binding, determined using either excess non-radioactive ligand or heat inactivation of the protein, as previously described (*Feng et al., 2015*). The TRPV1-activating double-knot toxin (DkTx) was produced recombinantly, folded in vitro and purified as previously described (*Bae et al., 2012*). Yeast membranes were incubated with DkTx solution for 1 hr at 4°C with gentle agitation prior to undertaking RTx binding assays.

## Electrophysiological recording of TPRV channels expressed in Xenopus oocytes

The WT rat TRPV1 and TRPV2 channels in the pcDNA3.1 vector were kindly provided by Dr. David Julius (UCSF) (*Caterina et al., 1997*; *Caterina et al., 1999*) and subcloned into the pGEM-HE vector (*Liman et al., 1992*). Mutations were introduced into rTRPV1 using a two-step PCR mutagenesis technique and the resulting constructs were verified by sequencing. All channel constructs were expressed in *Xenopus* oocytes and studied following 1–4 days incubation after cRNA injection (incubated at 17°C in 96 mM NaCl, 2 mM KCl, 5 mM HEPES, 1 mM $MgCl_2$ and 1.8 mM $CaCl_2$, 50 μg/ml gentamycin, pH 7.6 with NaOH) using the two-electrode voltage-clamp recording technique (OC-725C, Warner Instruments, Hamden, CT) with a 150 μl recording chamber. Data were filtered at 1–3 kHz and digitized at 20 kHz using pClamp software (Molecular Devices, Sunnyvale, CA). Microelectrode resistances were 0.1–1 MΩ when filled with 3 M KCl. For recording macroscopic TRP channel currents, the external recording solution contained (in mM): 100 KCl, 10 HEPES, pH 7.6 with KOH. All experiments were performed at room temperature (~22°C).

## Single channel recording of TRPV2 QM expressed in mammalian cells

HEK293 cells were cultured following standard protocols (*Li et al., 2015*) and transiently transfected with plasmids for the expression of TRPV2-QM (pcDNA1) and GFP (pGreen-Lantern, Invitrogen, Carlsbad, CA) using FuGENE6 (Promega, Madison, WI).

Standard whole-cell patch clamp recordings from transiently transfected HEK293 cells at room temperature (22–24°C) were performed. Data was acquired with an Axopatch 200B amplifier

(Molecular Devices, Sunnyvale, CA), filtered with an 8-pole low-pass Bessel filter (model 900, Frequency Devices, Ottawa, IL) and digitized with a Digidata 1550A interface and pClamp10 software (Molecular Devices). All data was analyzed using Igor Pro 6.34A (Wavemetrics Inc., Tigard, OR). Pipettes were pulled from borosilicate glass, covered in dental wax and heat-polished to final resistances between $10 - 15$ M$\Omega$. The intracellular recording solution consisted of (in mM): 130 NaCl, 10 HEPES, 10 EGTA, pH 7.4 (NaOH/HCl), and the extracellular solution (i.e. pipette solution) was supplemented with 10 mM $MgCl_2$. A gravity-fed rapid solution exchange system (RSC-200, BioLogic, Claix, France) was used. Excised patches were placed in front of glass capillaries perfused with different solutions.

Data were acquired at 20 kHz and low-pass filtered at 5 kHz, with an additional 2 kHz low-pass filter applied off-line. Single channel currents were recorded using a protocol consisting of 100 ms at a holding potential of $-90$ mV, 500 ms pulse at +90 mV, followed by another 100 ms at $-90$ mV, which was applied every 700 ms for the duration of the recording. Data collected at +90 mV and the 100 ms before the step to +90 mV were used for analyzing channel behavior at the two potentials.

A series of null-traces were obtained in control solution (in the absence of agonists) before application of RTx, and were used for subtraction of the leak currents and the capacitive transients. No spontaneous channel activity was recorded in control solution (data not shown). Recordings in RTx were started as soon as the patch was perfused with the agonist, which was followed by a 10–30 s delay period in which no channel activity was observed, followed by robust channel activation. To reflect the $P_o$ of RTx-bound channels, mean $P_o$ values shown in *Figure 6F* were calculated after the initial period where no channel activity was observed. The $P_o$ for every sweep was calculated using the 50% threshold crossing technique, which consists in generating idealized traces from the recordings, where data points above a specified threshold are assigned a value of 1 and data points below the threshold are assigned a value of 0, and the $P_o$ is calculated as the mean value of the idealized trace. For recordings at +90 mV, a threshold value of 4.5 pA was used (approximately half of the maximal conductance level of 10 pA), whereas a threshold of 1.2 pA was used for the recordings at -90 mV. A 5-pass binomial smoothing function was applied to the data at -90 mV prior to calculation of the $P_o$. For multi-channel patches, multiple thresholds were used and idealized traces were assigned a value of 2 when two channels were open simultaneously. For data obtained at +90 mV a threshold slightly lower than 50% of the maximal single-channel current amplitude was used so that subconductance levels were counted as channel openings. The mean $P_o$ data in *Figure 6F* is the average of the $P_o$-values for all sweeps in each patch, with an N = 2 for two-channel patches. For each patch the mean $P_o$ was also calculated from the normalized all-points histograms. In this case, histograms compiled from all recorded data-points after leak subtraction at a specified voltage were normalized to the area under the histogram curve. A Gaussian curve was fit to the data centered at 0 pA, and the area under its curve ($A_C$) was calculated. The $P_o$ was then calculated as $P_o = 1 - A_C$. Both methods for calculating $P_o$ gave identical results (data not shown). The single-channel current amplitudes were calculated from Gaussian fits to the all-points histogram portion corresponding to the current level of a single open channel.

## Acknowledgements

We thank Joe Mindell, Shai Silberberg, Gilman Toombes and members of the Swartz lab for helpful discussions. This work was supported by the Intramural Research Programs of the NINDS, NIH (to KJS) and NCI, NIH (to PMB; Z1A BC 005270), by a Welcome Trust Award (to SN; 102890/Z/13/Z), by an NINDS Competitive Postdoctoral Fellowship (to AJO), and by a grant from the KRIBB Research Initiative Program (Korean Biomedical Scientist Fellowship Program) of the Korea Research Institute of Bioscience and Biotechnology (to CB).

## Additional information

**Competing interests**
KJS: Reviewing editor, *eLife*. The other authors declare that no competing interests exist.

## Funding

| Funder | Grant reference number | Author |
|---|---|---|
| National Institute of Neurological Disorders and Stroke | NS002945 | Kenton J Swartz |
| National Cancer Institute | BC005270 | Peter M Blumberg |
| Wellcome Trust | 102890 | Simon Newstead |
| Korea Research Institute of Bioscience and Biotechnology | | Chanhyung Bae |
| National Institute of Neurological Disorders and Stroke | | Andres Jara-Oseguera |

The funders had no role in study design, data collection and interpretation, or the decision to submit the work for publication.

## Author contributions

FZ, Expressed 1D4-tagged TRPV1 and TRPV2, Made and studied all TRPV1 and TRPV2 mutants expressed in oocytes, Conception and design, Acquisition of data, Analysis and interpretation of data, Drafting or revising the article; SMH, Expressed and biochemically characterized all GFP-tagged TRPV1 constructs and performed RTx binding studies, Conception and design, Acquisition of data, Analysis and interpretation of data, Drafting or revising the article; AJ-O, Performed single channel recordings on TRPV2-QM, Conception and design, Acquisition of data, Analysis and interpretation of data, Drafting or revising the article; DK, Expressed and biochemically characterized all GFP-tagged TRPV1 constructs and performed RTx binding studies, Expressed 1D4-tagged TRPV1 and TRPV2, Conception and design, Acquisition of data, Analysis and interpretation of data, Drafting or revising the article; CB, Produced recombinant DkTx, Conception and design, Acquisition of data, Analysis and interpretation of data, Drafting or revising the article; LVP, Expressed and biochemically characterized all GFP-tagged TRPV1 constructs and performed RTx binding studies, Acquisition of data, Analysis and interpretation of data; PMB, SN, Conception and design, Analysis and interpretation of data, Drafting or revising the article; KJS, Conception and design, Analysis and interpretation of data, Drafting or revising the article, Contributed unpublished essential data or reagents

## Author ORCIDs

Sonya M Hanson, http://orcid.org/0000-0001-8960-5353
Simon Newstead, http://orcid.org/0000-0001-7432-2270
Kenton J Swartz, http://orcid.org/0000-0003-3419-0765

## Ethics

Animal experimentation: This study was performed in strict accordance with the recommendations in the Guide for the Care and Use of Laboratory Animals of the National Institutes of Health. All of the animals were handled according to approved institutional animal care and use committee (IACUC) protocol (#1253-15) of the National Institute of Neurological Disorders and Stroke.

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
