## [Decision Letter]

Thank you for submitting your article "Engineering vanilloid-sensitivity into the TRPV2 channel" for consideration by *eLife*. Your article has been reviewed by Rachelle Gaudet and Baron Chanda, and the evaluation has been overseen by a Reviewing Editor and Richard Aldrich as the Senior Editor.

The reviewers have discussed the reviews with one another and the Reviewing Editor has drafted this decision to help you prepare a revised submission.

Summary:

Despite the availability of a new higher resolution structure of TRPV2, the gating mechanisms of TRPV2 remain largely unknown. Unlike TRPV1 channels, vanilloid compounds do not activate TRPV2 channels, although both channels have a similar architecture. Here, the authors have used biochemical and electrophysiological approaches to identify the minimal determinants of resiniferatoxin (RTx) binding site in TRPV1 channels. Strikingly, they were able to transfer this binding site from TRPV1 to TRPV2 channels with as few as four binding site mutations, which also permitted investigation of the gating properties of TRPV2. This demonstrates that TRPV1 and TRPV2 not only share very similar structures, as demonstrated by recent cryoEM structures, but also very similar channel gating mechanics, and that a "quiescent" RTx binding pocket is present in TRPV2. This is a foundational study, which demonstrates that gating mechanism is conserved between these channels and the ligand sensing can be ported between various channels of this family.

Essential revisions:

The reviewers found the study to be of great interest and well executed, saying "Overall, the data are of high quality, well-illustrated and assessed with good statistics. The manuscript is very clearly written and the interpretations generally warranted. The results are of high importance to the field of ion channels, but also to those interested in pain sensing and sensory perception in general" and "I was impressed by the use of extensive biochemistry to demonstrate binding rather than limiting the study to dose response curves which is a standard in the field. Their binding measurements provide crucial evidence that the activation of the channels involve direct binding of RTx to TRPV1 and mutant TRPV2 channels. From a mechanistic standpoint, this is much more insightful because a dose-response curve would have just demonstrated that the activity is portable. Dose-response curves do not discriminate between the possibility that the RTx is also binding to TRPV2 but fails to activate the channel due to a lack of "proper" gating machinery or that the RTx just does not bind to the TRPV2 channel. In this study, they have addressed these questions comprehensively." The essential points to be addressed are the following:

1) Addressing the question of oligomerization either by correcting/editing the Discussion or by adding information about FSEC experiments, as described in the first point below.

2) Addressing the question of whether RTx may still bind the QM mutant, as described in the second point below, either experimentally with biochemical studies or through a clear discussion that clarifies and considers that possibility.

Detailed description for essential revisions:

1) I have one somewhat substantial concern, regarding the authors' discussions of oligomeric states: In the subsection “Allosteric interactions between DkTx and RTx”, the authors discuss possible oligomeric states for their truncated TRPV1 constructs. The authors seem to confuse the TRPA1 structure with that of TRPV1? A coiled coil was observed in the C-terminal region of the TRPA1 structure, not TRPV1. Also the TRPV1 structure suggests that any construct containing S5-S6 would likely need to tetramerize to form a native structure. Furthermore, SEC is a rather poor method, without additional information from other experiments, to assess oligomeric state. This comes up again in the Discussion, where the authors discuss SEC data that are not presented in the paper (only data for S1-S4 is shown, and S1-S6 and S1-Cterm constructs are discussed). It would be preferable to either include the data under Discussion and/or to tone down the arguments about oligomeric state, which are not particularly essential to the conclusions. If the authors want to include the current discussion of likely oligomeric state, then they should also include additional information about their FSEC experiments, such as the behavior of molecular weight standards on their column, and/or of related or previously validated constructs.

2) Although TRPV1 QM mutant does not clearly get activated by RTx, there is no accompanying biochemical evidence to show that these channels are not binding to RTx. The authors have shown that TRPV2QM is binding to RTx but there are other differences between TRPV1 and V2. Therefore, it is still possible that RTx is binding to TRPV1QM mutant but is unable to activate the channels.

---

## [Author Response]

Essential revisions:

The reviewers found the study to be of great interest and well executed, saying "Overall, the data are of high quality, well-illustrated and assessed with good statistics. The manuscript is very clearly written and the interpretations generally warranted. The results are of high importance to the field of ion channels, but also to those interested in pain sensing and sensory perception in general" and "I was impressed by the use of extensive biochemistry to demonstrate binding rather than limiting the study to dose response curves which is a standard in the field. Their binding measurements provide crucial evidence that the activation of the channels involve direct binding of RTx to TRPV1 and mutant TRPV2 channels. From a mechanistic standpoint, this is much more insightful because a dose-response curve would have just demonstrated that the activity is portable. Dose-response curves do not discriminate between the possibility that the RTx is also binding to TRPV2 but fails to activate the channel due to a lack of "proper" gating machinery or that the RTx just does not bind to the TRPV2 channel. In this study, they have addressed these questions comprehensively." The essential points to be addressed are the following:

1) Addressing the question of oligomerization either by correcting/editing the Discussion or by adding information about FSEC experiments, as described in the first point below.

2) Addressing the question of whether RTx may still bind the QM mutant, as described in the second point below, either experimentally with biochemical studies or through a clear discussion that clarifies and considers that possibility.

Detailed description for essential revisions:

1) I have one somewhat substantial concern, regarding the authors' discussions of oligomeric states: In the subsection “Allosteric interactions between DkTx and RTx”, the authors discuss possible oligomeric states for their truncated TRPV1 constructs. The authors seem to confuse the TRPA1 structure with that of TRPV1? A coiled coil was observed in the C-terminal region of the TRPA1 structure, not TRPV1. Also the TRPV1 structure suggests that any construct containing S5-S6 would likely need to tetramerize to form a native structure. Furthermore, SEC is a rather poor method, without additional information from other experiments, to assess oligomeric state. This comes up again in the Discussion, where the authors discuss SEC data that are not presented in the paper (only data for S1-S4 is shown, and S1-S6 and S1-Cterm constructs are discussed). It would be preferable to either include the data under Discussion and/or to tone down the arguments about oligomeric state, which are not particularly essential to the conclusions. If the authors want to include the current discussion of likely oligomeric state, then they should also include additional information about their FSEC experiments, such as the behavior of molecular weight standards on their column, and/or of related or previously validated constructs.

We agree with this point and have revised the manuscript to remove any mention of the oligomeric state from the Results section and only comment briefly in the Discussion (first paragraph) that the lack of binding of RTx to various deletion constructs could be explained if they are monomeric and therefore lacking the type of intersubunit binding pocket predicted by the cryo-EM structure.

*2) Although TRPV1 QM mutant does not clearly get activated by RTx, there is no accompanying biochemical evidence to show that these channels are not binding to RTx. The authors have shown that TRPV2QM is binding to RTx but there are other differences between TRPV1 and V2. Therefore, it is still possible that RTx is binding to TRPV1QM mutant but is unable to activate the channels.*

We have not tried to measure the binding of TRPV1-QM and decided against doing this because we would have to remake the mutants in the yeast expression construct and undertake several rounds of binding assays to optimize conditions since its likely that the expression level of the mutant would be different than our wt construct. We now comment on this in the Results (subsection “Interrogating the vanilloid binding pocket in TRPV1”, last paragraph) and point out that a previous study has shown that two of the four mutants in TRPV1-QM dramatically diminish binding of RTx, making it likely that the TRPV1-QM would bind RTx with very low affinity.